# Recognition of Long-COVID-19 Patients in a Canadian Tertiary Hospital Setting: A Retrospective Analysis of Their Clinical and Laboratory Characteristics

**DOI:** 10.3390/pathogens10101246

**Published:** 2021-09-26

**Authors:** Robert Kozak, Susan M. Armstrong, Elsa Salvant, Claudia Ritzker, Jordan Feld, Mia J. Biondi, Hubert Tsui

**Affiliations:** 1Sunnybrook Health Sciences Centre, Biological Science Platform, Sunnybrook Research Institute, Toronto, ON M4N 3M5, Canada; rob.kozak@sunnybrook.ca (R.K.); susan.armstrong@mail.utoronto.ca (S.M.A.); elsa.salvant@sri.utoronto.ca (E.S.); Claudia.ritzker@mail.utoronto.ca (C.R.); 2Toronto Centre for Liver Disease, University Health Network, Toronto, ON M5G 2C4, Canada; jordan.feld@uhn.ca (J.F.); mia.biondi@mail.mcgill.ca (M.J.B.)

**Keywords:** SARS-CoV-2, long-COVID, long-haulers, disease biomarkers, retrospective analysis

## Abstract

A proportion of patients with COVID-19 have symptoms past the acute disease phase, which may affect quality of life. It is important for clinicians to be aware of this “long-COVID-19” syndrome to better diagnose, treat, and prevent it. We reviewed clinical and laboratory characteristics of a COVID-19 cohort in a Toronto, Ontario tertiary care center. Demographic, clinical, and laboratory data were collected, and patients were classified as “long-COVID-19” or “non-long-COVID-19” using consensus criteria. Of 397 patients who tested positive for COVID-19, 223 met inclusion criteria, and 62 (27%) had long-COVID-19. These patients had a similar age distribution compared to non-long-COVID-19 patients overall but were younger in the admitted long COVID-19 group. The long-COVID-19 group had more inpatients compared to the non-long-COVID-19 group (39% vs. 25%) and more frequent supplemental oxygen or mechanical ventilation use. However, long-COVID-19 patients did not differ by duration of mechanical ventilation, length of stay, comorbidities, or values of common laboratory tests ordered. The most frequent symptoms associated with long-COVID-19 were fatigue and weakness, as reported most commonly by the infectious disease, respirology and cardiology disciplines. In conclusion, by retrospective chart review, 27% of COVID-19 patients presenting to a tertiary care center in Toronto, Canada, were found to meet criteria for long-COVID-19. Past medical history and routine laboratory testing at presentation did not predict for long-COVID-19 development.

## 1. Introduction

The COVID-19 pandemic has resulted in over 1.4 million cases and 26,000 deaths in Canada. Unfortunately, these numbers continue to rise. A proportion of SARS-CoV-2 infected patients have persistent symptoms referred to as post-acute COVID syndrome or “long-COVID-19” [1,2,3]. The occurrence of post-infectious syndromes has been reported for other viral infections such as Ebola and Zika viruses [4,5]. However, for many Canadian healthcare practitioners, this may be their first exposure to post-viral syndromes. Given the lack of awareness, recognition of the long-term sequelae of COVID-19 is important for planning long-COVID-19 management.

Long-COVID-19 can affect patients with pre-existing comorbidities, those who experienced severe acute illness during their infection, as well as young healthy individuals and/or those with mild illness [2,6,7,8]. Symptoms of long-COVID-19 include those that are characteristic of COVID-19 infection, such as fatigue, concentration difficulties, memory loss, shortness of breath, tinnitus, tremors, tachycardia, palpitations, vertigo, joint pain, and/or a rash [1,8,9]. Studies in China and the United States using patient cohorts with varying levels of long-COVID-19 severity found post-COVID symptoms to be debilitating and greatly affect quality of life [7,10]. Although the incidence of long-COVID-19 in Canada is unknown, it ranges from 10–76% in other countries [1]. In Italy, one study noted that, in hospitalized patients, 55% had three or more lingering symptoms and 44% reported worsened quality of life at a mean follow-up of 60 days after symptoms onset [8]. The prevalence in younger patients can be as high as 30%, affecting those as young as 18 years of age [10]. The National Institute for Health and Care Excellence guidelines currently define long-COVID-19 as “signs and symptoms that develop during or following an infection consistent with COVID-19, and which continue for more than four weeks and are not explained by an alternative diagnosis.” Presently, there are no recommendations regarding laboratory testing or objective biomarkers for long-COVID-19 [1,8,9]. Thus, differentiating long-COVID-19 from the natural history of existing comorbidities may be challenging.

There is a paucity of data on long-COVID-19 in Canada. Here, we describe the characteristics of patients with long-COVID-19 identified through a retrospective review of the first wave of COVID-19 patients in early to mid-2020 at our tertiary academic center in Toronto, Ontario. This information may help guide unified strategies to support long-COVID-19 patients while knowledge of its pathogenesis and treatment continue to evolve.

## 2. Results

### 2.1. Patient Demographics

A data query from 1 January, 2020, to 8 June, 2020 identified 397 unique patients with positive COVID-19 results (Figure 1). In total, 223 patients (56%) met criteria for retrospective chart review, and the details of patient classification is listed in Table 1. Within this cohort, 27% (*n* = 62) met long-COVID-19 criteria as defined by two or more persistent symptoms greater than 90 days after a positive SARS-CoV-2 RT-PCR test.

The mean age (49.6 vs. 48.6 years) was similar between the long-COVID-19 and non-long-COVID-19 groups. As shown in Table 2, the long-COVID-19 group was 61% female compared to 49% in the non-long-COVID-19 group (*p* = 0.13), a trend which has been noted in other studies [7].

### 2.2. Admission Status and Oxygen Requirements

The long-COVID-19 cohort consisted of 61% (*n* = 38) outpatients and 39% (*n* = 24) admitted patients. This distribution was significantly different from 75% (*n* = 121) outpatients and 25% (n = 40) admitted patients in the non-long-COVID-19 cohort (*p* = 0.04). For long-COVID-19 patients who were admitted, 16.7% (4/24) required no supplementary oxygen, and 83.3% (20/24) received some level of oxygen support. The subgroup of long-COVID-19 patients who received mechanical ventilation was 45.8% (11/24) (Table 3). In the admitted non-long-COVID-19 group 52.5% (21/40) required no supplementary oxygen, 47.5% (19/40) received some level of oxygen support, and 12.5% (5/40) required mechanical ventilation. The proportion of patients requiring any form of oxygen (group D + E) or mechanical ventilation (group E) was significantly higher in the long-COVID-19 group (*p* = 0.007 and *p* = 0.006, respectively). The number of ventilator days did not significantly differ between category E groups (mean 17.6 vs. 13.4 days, respectively, *p* = 0.53).

Whether the long-COVID-19 group represented a cohort that presented later to medical attention was analyzed by time from symptom onset to positive diagnostic result. This did not differ between long-COVID-19 and non-long-COVID-19 groups (mean 5.8 days vs. 5.0 days, *p* = 0.45), as shown in Table 2. Additionally, there was no difference from time of symptom onset to hospital admission (mean 7.9 days vs. 6.0 days, *p* = 0.11). Admitted patients meeting criteria for long-COVID-19 were also not admitted for longer than their non-long-COVID-19 counterparts (mean 26.4 days, range 4–163 vs. mean 11.2 days, range 1–60, *p* = 0.09), although it was noted that two long-COVID-19 patients were still admitted >90 days after COVID-19 diagnosis.

### 2.3. Comorbidities

The average number of comorbidities (2.2 vs 2.4 per patient, *p* = 0.79) was similar between admitted and outpatient long-COVID-19 and non-long-COVID-19 groups. There was no single comorbidity that was significantly associated with long-COVID-19 development, despite some enrichment in respiratory ailments (21% vs. 13.7%) and diabetes (19.4% vs. 11.2%; *p* = 0.21 and 0.12, respectively). The average number of comorbidities between admitted patients in each group (3.5 vs. 5.0 per patient) was compared and found not to be significantly different (*p* = 0.11). Subsequent analysis of admitted patients in both groups failed to identify any statistically significant differences in percentage of patients with a given comorbidity subtype.

### 2.4. Initial Laboratory Data

We examined routine laboratory results closest to the time of COVID-19 diagnosis to determine if early hematological or biochemical disturbances correlated with subsequent development of long-COVID-19. The LIS query generated a broad range of laboratory testing with the most frequently performed tests listed in Appendix A. The most common tests were complete blood count (CBC), creatinine, electrolytes (sodium, potassium, chloride), and blood gas. Given previous findings of CBC abnormalities related to severity of COVID-19, we limited further statistical analyses to CBC parameters and renal function. Basic coagulation testing was infrequently performed in the peri 7-day period from diagnosis. To investigate specific relationships with coagulopathy and long COVID-19, we analyzed the first available D-dimer result and found no statistical differences (Appendix A). Markers of inflammation were also rarely performed precluding meaningful analysis (e.g., ESR, LDH, CRP) (data not shown). 

There were no statistically significant differences in the core CBC elements of hemoglobin (*p* = 0.74), white blood cell count (*p* = 0.78), or platelets (*p* = 0.59) between long-COVID-19 and non-long COVID-19. Although D-dimer was not one of the most common tests performed in the peri-7-day interval around COVID-19 diagnosis (24 available results), given its potential association with chronic coagulopathy in long-COVID-1914, we expanded the search to any D-dimer results available. This revealed 24 results in the long-COVID-19 and non-long-COVID-19 groups. There was no statistical difference between D-dimer results between groups.

### 2.5. Data from Follow-Up Visits

To probe awareness of long-term complications of COVID-19 we explored which medical specialties cited long term complications > 90 days following COVID-19 diagnosis. The majority of clinical reports were generated by the infectious diseases service, followed by respirology and cardiology (Figure 2).

Additionally, we summarized the ongoing symptoms reported. Fatigue/weakness was the most common, followed by anxiety and then shortness of breath. However, a diversity of symptoms involving a range of systems were noted for all groups (Table 4A,B). The average number of symptoms reported was 4.5 (range 1–14).

## 3. Discussion

Post-acute sequelae from COVID-19 are increasingly recognized as a significant cause of morbidity [9,11]. Our data represent a descriptive analysis of long-COVID-19 in an urban Canadian cohort and showed that 28% of our cohort had long-COVID-19, with this condition being observed more often in admitted patients. These findings highlight the persistent effects of COVID-19 affecting both outpatient and inpatient cohorts including a large number of patients that managed their acute phase at home.

Our findings are similar to what has been described by other studies, although the incidence varies, ranging from 13% to 75%, likely in part due to differing diagnostic criteria and follow-up time [1,7,10,11,12]. Recently, a study from the US Department of Veterans Affairs highlighted the ongoing cost of COVID-19 as they reported that COVID-19 patients were at higher risk for ongoing symptoms, laboratory abnormalities, and death 6 months after their initial infection when compared to patients who had influenza [9]. Moreover, there was significant healthcare utilization by these individuals.

Interestingly, we observed in our cohort that 61% of long-COVID-19 patients were women, which is similar to recent studies in China and the United Kingdom that reported women are more likely to experience symptoms 6 months or greater than 1 month after infection, respectively [7,12]. However, in our study there were no significant differences in age, comorbidities, white blood cell count, or differential cell count. This suggests that existing health conditions and routine biochemistry/hematology laboratory testing at diagnosis are not sufficient to predict long-COVID-19 development.

There was a higher proportion of patients in the long-COVID-19 group who required oxygen or mechanical ventilation. The enrichment in exposure to supplemental oxygen in long-COVID-19 may be partially related to the definition of long-COVID-19 which is defined by symptoms and duration post COVID-19 diagnosis rather than objective laboratory or imaging findings. As such, there may be some overlap with post-ICU syndrome and long-COVID-19 acknowledging that the two are not necessarily mutually exclusive. As we obtain greater pathobiological understanding of long-COVID-19, it may be possible to better define or further subclassify patients experiencing post-COVID-19 clinical sequelae. However, the number of ventilator days did not significantly differ between category E groups suggesting length of ICU-admission was not the only driver of chronic post-COVID-19 symptoms in this cohort. Additionally, analysis of >90 day follow up visits indicated that while multiple specialists were involved in care, the majority of follow up was performed by infectious disease specialists. A systematic approach to querying patients for long COVID-19 symptoms may be of value where distinguishing long COVID-19 manifestations versus the natural history of pre-existing conditions may be important in future clinical management. Continuity of care for long-COVID patients, especially with community health care providers, may be an area of unmet need.

Our data included patients with documented clinical encounters at Sunnybrook Health Sciences Centre during the first wave of the pandemic, and thus only represents the demographics of our catchment area. The identification of long-COVID-19 syndrome was also dependent on medical record abstraction triggered by a subsequent medical interaction at Sunnybrook rather than a preplanned follow-up that specifically gauged symptoms associated with long-COVID-19. As such, a portion of patients who exhibited long-COVID-19 symptoms but either did not directly disclose them during a routine clinic visit, were overlooked during clinical assessment, were investigated elsewhere, or did not seek medical help were not captured by our approach. As such, this may contribute to a lower incidence of long-COVID-19 compared to other studies. Additionally, this may have contributed to small sample size observed in certain patient categories. These limitations highlight a major area for education where patients may have post-COVID sequelae that are not formally recognized during subspecialty follow-up. While laboratory utilization may be more consistent and routine at present, during the first wave we noted large variation in work-up, likely due to the limited formalized clinical guidance. Moreover, we did not collect socioeconomic, ethnicity, or mental health data, all of which may influence the development of long-COVID-19 syndrome. Our review of our cohort indicated only 1.3% (*n* = 3) had additional respiratory specimens collected for microbiological investigation, and no significant organisms were isolated from these samples. Routine testing for Epstein Barr Virus, or Cytomegalovirus, was not performed, but should be incorporated this into the diagnostic work up of future studies in order to better understand any potential contribution to the pathogenesis of long-COVID-19. Additionally, it cannot be definitively ruled out that other diseases may have contributed to persisting symptoms, although from our assessment, it would seem unlikely that other diagnoses would be the sole cause for presenting symptoms.

## 4. Materials and Methods

### 4.1. Identification and Characterization of Cohort

This retrospective study was performed on COVID-19 patients at Sunnybrook Health Sciences Centre (Toronto, Ontario, Canada), between 1 January 2020 and 8 June 2020, which coincided with the first wave in Ontario. These patients were identified through a Laboratory Information Systems (LIS) query, which was also used to obtain laboratory results. Patient demographics and clinical data were collected from the Sunnybrook electronic health record. Data related to persistent symptoms was reviewed up to 18 March 2021. Inclusion criteria were as follows: positive COVID-19 RT-PCR diagnostic result and presence of clinical follow-up as evidenced by clinical reports in the Sunnybrook electronic patient record. Patients were excluded if there was no or insufficient clinical data related to COVID-19 available in the Sunnybrook electronic health record (*n* = 142) or admission was due solely to disposition issues (*n* = 5). Such patients represented COVID-19 results for external clients which were processed through the Sunnybrook laboratory. Additionally, six patients were excluded due to age < 18 years.

Long-COVID-19 was defined as persistent symptoms greater than 90 days after initial SARS-CoV-2 detection [13]. Persistent symptoms were identified by chart review and identification was therefore limited to patients with clinical notes after 90 days from COVID-19 diagnosis or within 90 days if it was noted that their symptoms had completely resolved during that period. All COVID-19 patients were grouped [14], on the basis of admission status and oxygen requirements. Non-admitted patients were split into those with and without blood-based laboratory results. Category A included non-admitted patients with no bloodwork performed, Category B included non-admitted patients with bloodwork, Category C included admitted patients who did not require supplemental oxygen, Category D included admitted patients who required supplemental oxygen, but not mechanical ventilation, and Category E included admitted patients who required mechanical ventilation at any point during admission.

Patient comorbidities were those identified at the time they tested positive for COVID-19 and were subsequently arranged into organ systems, including cardiovascular (e.g., atrial fibrillation, myocardial infarction, hypertension, congestive heart failure, peripheral vascular disease, stroke or transient ischemic attack, coronary artery bypass graft, and/or dyslipidemia), respiratory/lung (e.g., asthma, chronic obstructive pulmonary disease, interstitial lung disease, obstructive sleep apnea, and/or pneumonia within last 12 months), diabetes (e.g., type 1 or 2), chronic renal disease (variable etiology), or other [15]. Total comorbidities were based on the number of specific comorbidities, not the number of organ systems affected. Laboratory data analyzed were limited to the value closest to the date of the COVID-19 swab within a +/− 7-day interval.

### 4.2. Statistical Analysis

Mean, standard deviation, and percentages as well as unpaired *t*-tests, Welch’s *t*-test, and Fisher’s exact test for testing statistical significance where appropriate were calculated using GraphPad Prism Version 9.1.1.225 (https://www.graphpad.com/scientific-software/prism/ accessed on 21 September 2021).

### 4.3. Ethics Approval

The Sunnybrook Health Sciences Centre Research Ethics Review Board approved this study (REB #1911).

## 5. Conclusions

Our findings highlight the need for more objective diagnostic modalities beyond clinical history and conventional laboratory tests in order to better identify cases and characterize this disease. Additionally, the role of SARS-CoV-2 variants of concern in the pathogenesis of long-COVID-19 syndrome should be a goal of future studies. SARS-CoV-2 has been shown to target multiple organs [16,17], potentially explaining the constellation of symptoms seen in this disease, patients may seek care from a range of specialists or practitioners depending on how their post-acute sequelae present. This highlights an important continuing educational need for healthcare providers in all disciplines in this area and underscores the need for development of clear criteria for case identification.

## Figures and Tables

**Figure 1 pathogens-10-01246-f001:**
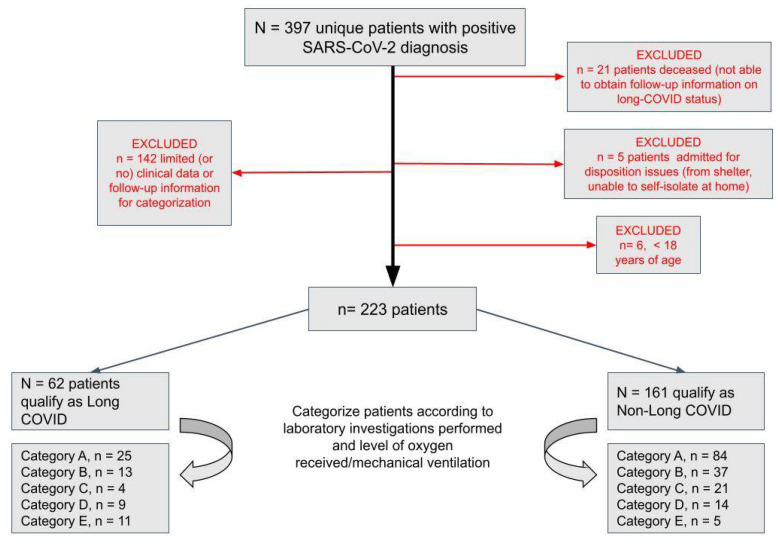
Flowchart of patients diagnosed with SARS-CoV-2 between 1 January and 8 June 2020. Patient cohorts characterized according to laboratory investigations performed and level of oxygen received/mechanical ventilation.

**Figure 2 pathogens-10-01246-f002:**
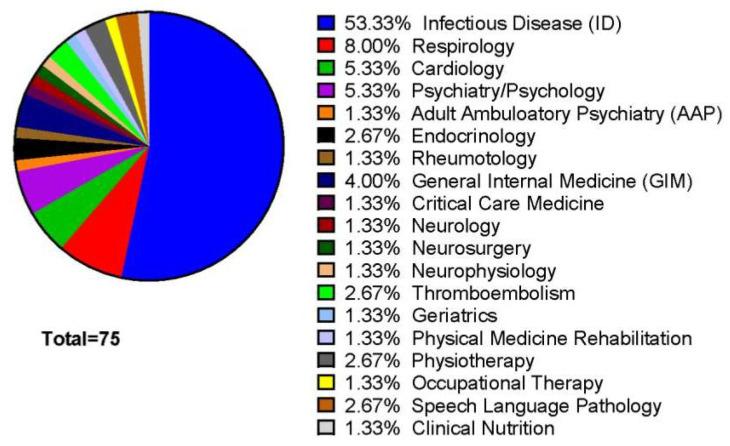
Follow-up visits amongst long-COVID-19 cohort. Appointments with various specialty services 3 months or greater after the COVID-19 diagnosis are listed.

**Table 1 pathogens-10-01246-t001:** Patient Category Criteria.

Patient Category	Criteria
Category A	Non-admitted patients presented to hospital with no bloodwork performed
Category B	Non-admitted patients presented to hospital with bloodwork performed
Category C	admitted to hospital and not requiring any supplemental oxygen while admitted
Category D	admitted to hospital and only required supplemental oxygen while admitted
Category E	admitted to hospital and required mechanical ventilation at any point during admission

Patients classified according to their highest stratified category during their stay.

**Table 2 pathogens-10-01246-t002:** Long-COVID and Non-Long-COVID cohort characteristics.

	Long COVID Group, *n* = 62						Non-Long COVID Group, *n* = 161						*p*-Value
	**Total**	**A**	**B**	**C**	**D**	**E**	**Total**	**A**	**B**	**C**	**D**	**E**	
Age, mean (sd){range} [confidence int]	49.1 (14.1)	42.9 (12.0)	49.2 (11.1)	58.8 (16.6)	54.3 (15.0)	55.5 (13.1)	48.6 (19.3)	39.2 (12.8)	52.0 (18.4)	64.7 (18.8)	71.4 (19.4)	51.4 (8.2)	0.8245
	23–78	23–66	30–67	32–76	30–77	23–78	19–97	19–66	19–89	21–94	18–97	37–59	
	45.6 to 52.6	38.2 to 47.6	43.2 to 55.3	42.5 to 75.0	44.6 to 64.1	47.7 to 63.2	45.6 to 51.6	36.4 to 41.9	46.1 to 57.9	56.7 to 72.7	61.3 to 81.6	44.2 to	
Male, No (%)	24 (38.7)	7 (28.0)	6 (46.2)	1 (25.0)	5 (55.6)	5 (45.5)	81 (50.3)	44 (52.4)	16 (43.2)	11 (52.4)	6 (42.9)	4 (80.0)	0.1357
Female, No (%)	38 (61.3)	18 (72.0)	7 (53.8)	3 (75.0)	4 (44.4)	6 (54.5)	80 (49.7)	40 (47.6)	21 (56.8)	10 (47.6)	8 (57.1)	1 (20.0)	
Comorbidities													
Cardiovascular, No (%)	24 (38.7)	5 (20.0)	6 (46.2)	2 (50.0)	3 (33.3)	8 (72.7)	56 (34.8)	12 (14.3)	18 (48.6)	17 (81.0)	8 (57.1)	1 (20.0)	0.6409
Respiratory/Lung, No (%)	13 (21.0)	3 (12.0)	1 (7.7)	3 (75.0)	1 (11.1)	5 (45.5)	22 (13.7)	9 (10.7)	6 (16.2)	5 (23.8)	2 (14.3)	0 (0.0)	0.2174
Diabetes, No (%)	12 (19.4)	2 (8.0)	1 (7.7)	2 (50.0)	2 (22.2)	5 (45.5)	18 (11.2)	5 (6.0)	5 (13.5)	5 (23.8)	3 (21.4)	0 (0.0)	0.1264
Chronic Renal Disease No (%)	3 (4.8)	0 (0.0)	0 (0.0)	0 (0.0)	1 (11.1)	2 (18.2)	11 (6.8)	1 (1.2)	4 (10.8)	3 (14.3)	2 (14.3)	1 (20.0)	0.7622
Total Comorbidities Mean (sd) {range} [confidence int]	2.2 (2.7)	1.4 (1.5)	1.5 (1.4)	4.5 (3.6)	1.9 (2.3)	4.4 (3.9)	2.4 (3.0)	0.90 (1.2)	2.8 (2.9)	6.0 (4.0)	4.5 (2.8)	2.0 (1.4)	0.7955
	0–12	0–6	0–4	0–8	0–8	0–12	0–16	0–5	0–14	1–16	0–10	0–4	
	1.6 to 2.9	0.84 to 2.0	0.78 to 2.3	1.0 to 8.0	0.40 to 3.4	2.1 to 6.6	1.9 to 2.8	0.66 to 1.2	1.9 to 3.7	4.3 to 7.7	3.0 to 6.0	0.76 to 3.2	

Values shown are mean (standard deviation), (No) number and proportion of total, range if applicable, and confidence interval with α = 0.05. Statistical tests used were unpaired *t*-tests, unpaired *t*-tests with Welch’s corrections and Fisher’s exact test for comparing proportions.

**Table 3 pathogens-10-01246-t003:** Admission and diagnosis characteristics.

	Long COVID Group, *n* = 62						Non-Long COVID Group, *n* = 161						*p*-Value
	**Total**	**A**	**B**	**C**	**D**	**E**	**Total**	**A**	**B**	**C**	**D**	**E**	
Times tested for COVID, Mean (SD){range} [confidence int]	2.3 (1.8)	1.8 (0.8)	1.4 (0.62)	2.5 (1.1)	1.7 (1.1)	4.7 (2.7)	2.0 (1.6)	1.7 (1.2)	0.18 (2.2)	2.9 (2.6)	2.4 (1.6)	2.8 (1.2)	0.2664
	1–9	1–3	1–3	1–4	1–4	1–9	1–10	1–7	(−5)–6	1–10	1–6	1–4	
	1.8 to 2.7	1.5 to 2.1	1.0 to 1.7	1.4 to 3.6	0.98 to 2.4	3.1 to 6.3	1.7 to 2.2	1.5 to 1.9	1.3 to 2.4	1.8 to 4.0	1.6 to 3.3	1.8 to 3.8	
Time between symptom onset and COVID + diagnosis (days), mean (sd) {range} [confidence int]	5.8 (6.7)	4.5 (7.2)	6.6 (8.9)	6.5 (3.4)	9.4 (1.9)	4.5 (4.4)	5.0 (6.8)	4.7 (6.9)	7.2 (7.8)	3.3 (6.2)	5.1 (4.4)	2.6 (1.5)	0.4549
	(−3)–36	(−3) to 34	0–36	1–10	6–13	(−2)–14	(−9)–36	(−2) to 36	0–31	(−9)–18	0–13	0–4	
	4.1 to 7.5	1.5 to 7.4	1.8 to 11.5	3.2 to 9.8	8.2 to 10.7	2.0 to 7.1	3.9 to 6.2	3.1 to 6.3	4.5 to 9.9	0.46 to 6.2	2.8 to 7.4	1.3 to 3.9	
Time between COVID+ diagnosis and admission (days), Mean (sd) {range} [confidence int]	1.2 (3.0)	--	--	1.3 (4.1)	0.0 (1.2)	2.1 (3.1)	2.1 (2.8)	--	--	2.1 (2.8)	2.1 (2.7)	2.4 (3.1)	0.2333
	(−3)–8	--	--	(−3)–8	(−3)–2	(−2)–8	(−2)–9	--	--	0–9	(−2)–8	(−1)–7	
	−0.03 to 2.4	--	--	(−2.8) to 5.3	(−0.81) to 0.81	0.23 to 4.0	1.1 to 3.1	--	--	0.62 to 3.5	0.54 to 3.6	(−0.29) to 5.1	
Length of Admission (days), mean (sd) {range} [confidence int]	26.4 (41.0)	--	--	4.0 (0.0)	8.1 (4.3)	49.5 (51.6)	11.2 (11.0)	--	--	5.4 (4.6)	11.2 (4.3)	27.6 (17.0)	0.0965
	4–163	--	--	4–4	5–17	13–163	1–60	--	--	1–16	3–17	11–60	
	10.0 to 42.8	--	--	--	5.3 to 10.9	19.0 to 79.9	7.3 to 15.1	--	--	3.0 to 7.9	8.7 to 13.6	12.7 to 42.5	
Time between symptom onset and admission (days), Mean (sd) {range} [confidence int]	7.9 (3.3)	--	--	7.8 (0.83)	9.4 (1.9)	6.6 (4.2)	6.0 (4.7)	--	--	4.6 (5.3)	8.0 (3.7)	5.0 (2.6)	0.1159
	2–14	--	--	7–9	7–13	2–14	0–20	--	--	0–20	3–16	2–9	
	6.5 to 9.2	--	--	6.9 to 8.6	8.2 to 10.7	4.2 to 9.1	4.4 to 7.7	--	--	1.7 to 7.5	5.9 to 10.1	2.7 to 7.3	
Days on supplemental O2, mean (sd) {range} [confidence int]	--	--	--	--	5.2 (3.8)	21.0 (16.5)	--	--	--	--	8.7 (4.4)	20.25 (6.9)	0.6981
	--	--	--	--	1–12	11–66	--	--	--	--	1–15	15–32	
	--	--	--	--	2.7 to 7.7	10.2 to 31.8	--	--	--	--	6.2 to 11.2	13.5 to 27.0	
Days on ventilator, Mean (sd) {range} [confidence int]	--	--	--	--	--	17.6 (12.0)	--	--	--	--	--	13.4 (9.0)	0.5315
	--	--	--	--	--	8–49	--	--	--	--	--	3–30	
	--	--	--	--	--	10.2 to 25.0	--	--	--	--	--	5.5 to 21.3	

Values shown are mean (standard deviation), range if applicable and confidence interval with α = 0.05. Statistical tests used were unpaired *t*-tests, unpaired *t*-tests with Welch’s corrections and Fisher’s exact test for comparing proportions.

**Table 4 pathogens-10-01246-t004:** **A.** Symptoms in long-COVID-19 cohort ≥ 90 days. **B**. Persistent symptoms summarized per category in long-COVID cohort.

A	Number (Total)	%	Group A	%	Group B	%	Group C	%	Group D	%	Group E	%
SOB/SOBOE	17	27.419	7	28	2	15.385	2	50	1	11.11	5	50
Runny Nose/Congestion	7	11.290	4	16	1	7.692	0	0	1	11.11	1	10
Sore Throat	3	4.839	1	4	2	15.385	0	0	0	0.00	0	0
Dysphagia/Difficulty Swallowing	2	3.226	0	0	0	0	0	0	0	0.00	2	20
Cough	7	11.290	3	12	2	15.385	0	0	0	0.00	2	20
Wheezing	2	3.226	1	4	0	0	0	0	0	0.00	1	10
Dyspnea/DOE	12	19.355	5	20	1	7.692	1	25	1	11.11	4	40
Sputum	6	9.677	2	8	0	0	1	25	1	11.11	2	20
Chest Pain/Tightness/Heaviness	15	24.194	4	16	5	38.462	0	0	2	22.22	4	40
Tachycardia/Palpitations	4	6.452	2	8	0	0	0	0	1	11.11	1	10
Fatigue/Weakness/Tiredness	31	50.000	11	44	8	61.538	3	75	5	55.56	4	40
Hair Loss/Thinning	22	35.484	7	28	4	30.769	1	25	5	55.56	5	50
Muscle Weakness	8	12.903	1	4	2	15.385	0	0	2	22.22	3	30
Myalgia	7	11.290	1	4	2	15.385	1	25	1	11.11	2	20
Arthralgia	6	9.677	1	4	3	23.077	1	25	0	0.00	1	10
Abdominal Pain/Nausea/Bowel Movements	4	6.452	1	4	2	15.385	0	0	0	0.00	1	10
Rash/Dermatological Issues	8	12.903	1	4	4	30.769	0	0	1	11.11	2	20
Anosmia	19	30.645	9	36	3	23.077	1	25	2	22.22	3	30
Dysgeusia	10	16.129	7	28	0	0	0	0	1	11.11	2	20
Ear Pain/Hearing Issues	4	6.452	1	4	4	30.769	0	0	0	0.00	0	0
Headache/Head Heaviness	12	19.355	3	12	4	30.769	1	25	0	0.00	4	40
Brain Fog/Cognitive Issues	14	22.581	2	8	4	30.769	2	50	1	11.11	5	50
Anxiety/PTSD	22	35.484	9	36	5	38.462	1	25	3	33.33	4	40
Depression	8	12.903	1	4	4	30.769	1	25	0	0.00	2	20
Anorexia/Decrease in appetite	3	4.839	0	0	2	15.385	0	0	1	11.11	0	0
Insomnia	14	22.581	4	16	2	15.385	1	25	3	33.33	4	40
Adenopathy	1	1.613	0	0	1	7.692	0	0	0	0.00	0	0
Diplopia/Bilateral eye pain	2	3.226	0	0	2	15.385	0	0	0	0.00	0	0
Neurological Issues (Numbness/Tingling)	3	4.839	0	0	0	0	0	0	0	0.00	3	30
**B**	**Number of Persistent Symptoms**											
	**Total (*n* = 61)**		**Category A** **(*n* = 25)**		**Category B** **(*n* = 13)**		**Category C** **(*n* = 4)**		**Category D** **(*n* = 9)**		**Category E** **(*n* = 10)**	
Mean (SD)	4.52 (3.08)		3.52 (1.72)		5.31 (3.95)		4.25 (3.49)		3.78 (2.25)		6.8 (3.4)	
Range	1 to 14		1 to 7		1 to 14		1 to 10		1 to 7		2 to 11	
IQR	4		3		3		3.75		4		6.75	

## Data Availability

Not applicable.

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
