# Peer review of "Recognition of Long-COVID-19 Patients in a Canadian Tertiary Hospital Setting: A Retrospective Analysis of Their Clinical and Laboratory Characteristics"

_pathogens, 2021, doi:10.3390/pathogens10101246_

Round 1
Reviewer 1 Report
This is an important and thought-provoking manuscript regarding long Covid consequences in Canada that deals with an unfamiliar phenomenon that has to be studied.
- I highly recommend moving the Methods section up, before the Results section for a better understanding of the whole manuscript.
- Please add a column of P value for each line of the various tables. That way the significance of results would be clearer.
- Table 3 is unnecessary, especially since there is no data of significance. The results are detailed in the text.
- Please add the relevant acronyms at the bottom of the tables.
- Since there was a difference in the oxygen or mechanical ventilation requirements between the groups, can you elaborate any explanation for this finding? Please discuss this finding.
- Were there any differences in the radiographic abnormalities between long Covid patients and others?
Author Response
1) I highly recommend moving the Methods section up, before the Results section for a better understanding of the whole manuscript.
We thank the reviewers for this suggestion and have made the changes to the manuscript to facilitate better understanding. Note, the methods section was placed at the end of the manuscript per the formatting template for this journal.
2) Please add a column of P value for each line of the various tables. That way the significance of results would be clearer.
We thank the reviewer for this comment and have modified the tables in the manuscript to also include a column with P values.
3) Table 3 is unnecessary, especially since there is no data of significance. The results are detailed in the text.
We appreciate the suggestion from the reviewer and have moved Table 3 to a supplemental table in order to still ensure that the data is available to the readership.
4) Please add the relevant acronyms at the bottom of the tables.
We have made these changes to the tables.
5) Since there was a difference in the oxygen or mechanical ventilation requirements between the groups, can you elaborate any explanation for this finding? Please discuss this finding.
We thank the reviewer for bringing this to our attention. The enrichment in exposure to supplemental oxygen in long-COVID19 may be partially related to the definition of long-COVID19 which is defined by symptoms and duration post COVID19 diagnosis rather than objective laboratory or imaging findings. As such, there may be some overlap with post-ICU syndrome and long-COVID19 acknowledging that the two are not necessarily mutually exclusive. As we obtain greater pathobiological understanding of long-COVID19, it may be possible to better define or further subclassify patients experiencing post-COVID19 clinical sequelae. We have added this to the discussion section of the manuscript (lines 193-200).
6) Were there any differences in the radiographic abnormalities between long Covid patients and others?
We thank the reviewer for the suggestion of integrating diagnostic imaging. Of the 223 patients in our cohort only 39% (n=88) had at least one diagnostic imaging modality on record within 14 days of their COVID-19 diagnosis. Additionally, more imaging was performed in the long COVID-19 group (n=31, 50%) which could bias findings. Moreover, these radiologic reports were descriptive and not readily amenable to quantitative assessment without scoring and review of the primary images by trained radiologists. This would be an interesting endeavor but beyond the scope of the current manuscript.
Reviewer 2 Report
Thank you for your submission, I hope you find my comments helpful.
It looks like you have left out the "Methods" section entirely, going straight from "1. Introduction" to "2. Results" section. This makes it impossible to understand your results section.
I am unable to interpret your tables. First of all, I do not see where you describe groups A, B, C, D, and E (there is no "methods" section). Secondly, there is so much data in each table that it becomes so confusing that I just gave up trying to figure out what you are getting at.
A manuscript is meant to summarize the data for readers. You are not publishing a book. You need to spend time really thinking about your findings. Identify important findings. Determine what findings are important, and what findings aren't important. Don't just do a data dump.
There is great value in providing some simplicity. I understand that you collected lots of data. Great. But in the manuscript you need to take that data and make sense of it. Don't just regurgitate all of your data in the tables embedded in the manuscript. You can provide a data supplement to the manuscript for all of your data in order to be complete, but the tables in the manuscript are meant to present your findings in a meaningful way that is understandable to a wide range of readers from varying backgrounds.
For example, a 21 x 12 table is really excessive (Table 2). Summarize your data. Simplify. Emphasize key points. You presented so much data, how can I look at your table and determine what is important? Is it row 13 column 7? Or is it row 19 column 10? Why make it nearly impossible for readers to identify what is important?
I simply cannot understand your manuscript because after looking several times, I still cannot find where you describe what the groups A, B, C, D, and E stand for other than the legend for Figure 1 where you say that the patient cohorts were characterized according to laboratory investigations and level of oxygen received/mechanical ventilation. This is an insufficient description of your groups.
Author Response
- It looks like you have left out the "Methods" section entirely, going straight from "1. Introduction" to "2. Results" section. This makes it impossible to understand your results section.
The methods section is included at the end of the paper per the formatting template of the journal.
2. I am unable to interpret your tables. First of all, I do not see where you describe groups A, B, C, D, and E (there is no "methods" section). Secondly, there is so much data in each table that it becomes so confusing that I just gave up trying to figure out what you are getting at. A manuscript is meant to summarize the data for readers. You are not publishing a book. You need to spend time really thinking about your findings. Identify important findings. Determine what findings are important, and what findings aren't important. Don't just do a data dump. There is great value in providing some simplicity. I understand that you collected lots of data. Great. But in the manuscript you need to take that data and make sense of it. Don't just regurgitate all of your data in the tables embedded in the manuscript. You can provide a data supplement to the manuscript for all of your data in order to be complete, but the tables in the manuscript are meant to present your findings in a meaningful way that is understandable to a wide range of readers from varying backgrounds. For example, a 21 x 12 table is really excessive (Table 2). Summarize your data. Simplify. Emphasize key points. You presented so much data, how can I look at your table and determine what is important? Is it row 13 column 7? Or is it row 19 column 10? Why make it nearly impossible for readers to identify what is important?
We appreciate the suggestion from the reviewer and have moved Table 3 to a supplemental table in order to still ensure that the data is available to the reader. The methods section is included towards the end of the manuscript as per the formating requirements of the journal.
3. I simply cannot understand your manuscript because after looking several times, I still cannot find where you describe what the groups A, B, C, D, and E stand for other than the legend for Figure 1 where you say that the patient cohorts were characterized according to laboratory investigations and level of oxygen received/mechanical ventilation. This is an insufficient description of your groups.
We have included a table in the methods that describes the categories of patients to make this readily accessible to the readership.
Reviewer 3 Report
Recognition of long-COVID-19 patients in a Canadian tertiary hospital setting: A retrospective analysis of their clinical and laboratory characteristics” by Kozak et al., describe the post-acute COVID syndrome or “long-COVID-19” which could affect the quality of life. It is necessary to understand the long-term effect of COVID-19, specifically with diabetes and respiratory ailments.
My concerns are listed below:
- Were the patients showing any secondary infections from other viruses or bacteria? Other pathogen’s role in exacerbating the pathogenicity of COVID-19 may be discussed.
- The category A-E may be explained in detail or provide a table for the criteria for classification.
- A summary of the study required.
- Please add the conclusion of the study in the abstract
- Fig 2, authors may look for the correlation between diseases onset and covid 19.
- Category C in long covid19, has only n=4, that could have merged with D or B, that low number is too difficult to conclude anything and is insufficient for stat analysis.
Author Response
1) Were the patients showing any secondary infections from other viruses or bacteria? Other pathogen’s role in exacerbating the pathogenicity of COVID-19 may be discussed.
We thank the reviewer for this point. Our review of the patients within this cohort indicated only 1.3% (n=3) had respiratory specimens collected, and no significant organisms were isolated from these samples. Additionally, no other respiratory viruses were detected in these patients, however it must be considered that not all patients underwent multiplex testing for additional respiratory viruses once the diagnosis of COVID-19 had been made. A limitation of our study is that patients were not routinely tested for Epstein Barr Virus, or Cytomegalovirus, both of which can exacerbate other infection when reactivated. Future studies should incorporate this into the diagnostic work up in order to better understand the pathogenesis of long-COVID-19. We have added to this to the discussion (lines 225-230)
2) The category A-E may be explained in detail or provide a table for the criteria for classification.
We thank the reviewer for this and we have included a table that describes the categories of patients.
3) A summary of the study required.
A summary has been included below the methods (section 5) as per the journal formatting requirements.
4) Please add the conclusion of the study in the abstract
We appreciate the suggestion from the reviewer and have added a conclusion sentence at the end of the abstract.
5) Fig 2, authors may look for the correlation between diseases onset and covid 19.
We appreciate the reviewers bringing this to our attention. Regrettably, due to the fact this was a retrospective review we were unable to review the symptomatology on a prospective basis. Thus, it would be challenging to draw firm conclusions between symptom onset and the likelihood of developing long-COVID. However, this is an interesting endeavor for a future study.
6) Category C in long covid19, has only n=4, that could have merged with D or B, that low number is too difficult to conclude anything and is insufficient for stat analysis.
We thank the reviewer for this suggestion and acknowledge that this is a small sample size for this group but this is actually what our retrospective review discovered. We believe it is appropriate to keep these individuals in this category, as they represent a relatively rare cohort of COVID19 infected patients who were deemed to require hospital admission but did not need any supplemental oxygen. This category is subject to clinical judgement and influenced by social supports as well as logistical determinants such as bedspace restrictions which we did not track. Given the fluid nature of hospital admissions and COVID19 ICU transfers to our tertiary centre, category C was anticipated to be lower. We have added to the discussion that low numbers in some categories are a potential limitation of our study (line 218-219). Note that outpatient (A+B) vs inpatient (C+D+E) analyses were included in the text and thus group C was indeed merged with D+E for some analyses (as per reviewer recommendation).
Round 2
Reviewer 1 Report
Thank you for the changes.
Good luck.
Reviewer 2 Report
Thank you for your revision and please accept my apologies for not seeing the methods section at the end of the article. Now it makes much more sense!
I think your tables are OK and acceptable. However, I do recommend that in future research you focus on making data presentation both complete and at the same time succinct and direct.
Thank you once again for all of your hard work and I hope you continue researching this important topic.
Reviewer 3 Report
The authors have answered the queries wih sufficient modifications of the text and tables.